# Quantitative analysis of *C. elegans* transcripts by Nanopore direct-cDNA sequencing reveals terminal hairpins in non *trans*-spliced mRNAs

Florian Bernard [1,2], Delphine Dargère[1], Oded Rechavi[2] & Denis Dupuy [1] ✉

In nematodes and kinetoplastids, mRNA processing involves a *trans*-splicing step through which a short sequence from a snRNP replaces the original 5' end of the primary transcript. It has long been held that 70% of *C. elegans* mRNAs are submitted to trans-splicing. Our recent work suggested that the mechanism is more pervasive but not fully captured by mainstream transcriptome sequencing methods. Here we use Oxford Nanopore's long-read amplification-free sequencing technology to perform a comprehensive analysis of *trans*-splicing in worms. We demonstrate that spliced leader (SL) sequences at the 5' end of the mRNAs affect library preparation and generate sequencing artefacts due to their self-complementarity. Consistent with our previous observations, we find evidence of *trans*-splicing for most genes. However, a subset of genes appears to be only marginally *trans*-spliced. These mRNAs all share the capacity to generate a 5' terminal hairpin structure mimicking the SL structure and offering a mechanistic explanation for their non conformity. Altogether, our data provide a comprehensive quantitative analysis of SL usage in *C. elegans*.

*Trans*-splicing is an RNA maturation process originally discovered in Trypanosomes, where gene expression is polycistronic and the resulting long RNAs need to be matured into monocistronic units before being fully functional. The first Spliced Leader sequence (SL) was detected during the characterization of cDNAs clones encoding surface glycoproteins (VSGs)[1]. It has then been shown that coupling of *trans*-splicing and polyadenylation is a crucial step of the maturation of the RNAs in Trypanosomes[2]. In these protists, genes do not contain introns, therefore the spliceosome's sole function is to perform *trans*-splicing so that all the mRNAs begin with a SL sequence of 39 nt derived from the 5' end of a 137-nt RNA called medRNA. After the discovery of *trans*-splicing in trypanosomes, it was found to exist in other metazoans, including cnidarians, ctenophores, rotifers,

flatworms, nematodes, crustaceans, and sponges. However, *trans*-splicing has not been found in any plants, fungi, insects or vertebrates[3].

In the nematode *Caenorhabditis elegans, trans*-splicing was initially uncovered during the study of the 5' extremity of the actin mRNA, which was found to contain an additional 22nt sequence that was absent from the genomic copy of the gene[4]. This first nematode SL sequence (SL1) was revealed to be donated by a 100-nt small nuclear ribonucleoprotein particle (snRNP). This process is closely related to *cis*-splicing (intron removal) where the 5' splice site is on the SL RNA, and the site of SL addition (3' splice site) is on the pre-mRNA[5]. The reaction happens through a branched intermediate, similar to the lariat of *cis*-splicing: cleavage of the SL sequence by the 5' region of the pre-mRNA in a first step, leading to the formation of a branched

[1]Université de Bordeaux, Inserm U1212, CNRS UMR5320, Institut Européen de Chimie et Biologie (IECB), 2, rue Robert Escarpit, 33607 Pessac, France.
[2]Department of Neurobiology, Wise Faculty of Life Sciences & Sagol School of Neuroscience, Tel Aviv University, Tel Aviv, Israel.
✉e-mail: denis.dupuy@inserm.fr

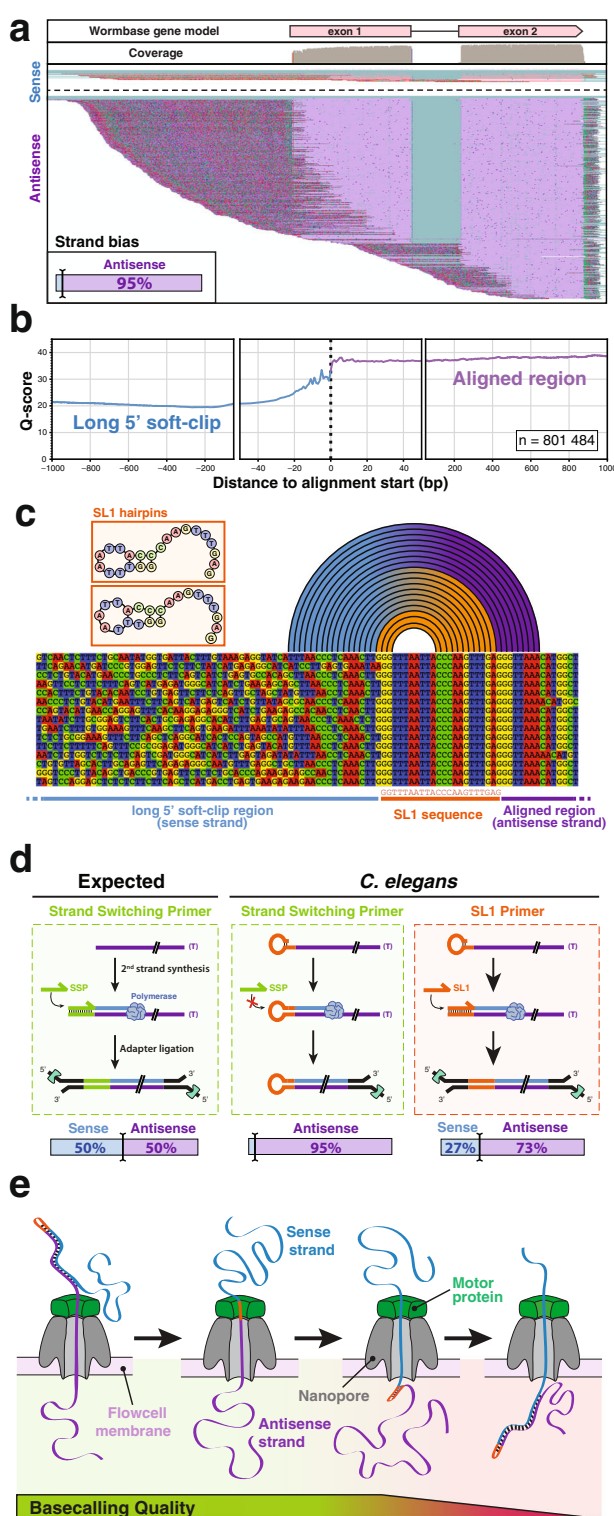

**Fig. 1 | Strand bias in Nanopore direct-cDNA sequencing of *C. elegans* transcripts. a** Typical genome browser (IGV) view of direct-cDNA reads aligned on a *C. elegans* gene (here gene *lys-1*). Aligned bases from the sense strand reads are shown in pink and aligned bases from the antisense strands in purple. Unaligned soft-clip region is shown with mismatches colored according to the observed base. Inset: overall strand bias observed across all detected transcripts. **b** Base quality measured in 5′ soft-clip and primary alignments. The average base-quality value over ~800,000 individual Nanopore reads was measured. **c** Typical alignment of reads at the interface between the transcript match and the long soft-clipped region. The first bases of the soft clip commonly correspond to the SL1 sequence followed by a partial antisense SL1 segment (blue to orange arcs) consistent with the extension of one of the endogenous hairpins represented as insets. The arcs indicate base pairing between the 5′ soft-clip and the *trans*-spliced antisense strand. **d** Schematic model of molecular events occurring during the second-strand synthesis step in various conditions of direct-cDNA sequencing library preparation. Bottom: observed strand bias measured after Minion sequencing. **e** Schematic representation of the progression of a hairpin cDNA substrate through the pore during Nanopore sequencing. The helicase activity of the motor protein maintains a steady rate of transfer for the antisense strand while unwinding the double-stranded cDNA. For the second strand, the absence of double helix on the input side and/or the kinetics of the base pairing on the other side of the pore affects the signal quality and prevents accurate base-calling of the sense strand.

that gene clusters act in a similar way as bacterial operons, with the transcription of the entire cluster being driven by a promoter situated on the 5′ extremity of the region. However, while in bacteria mRNAs from operons remain polycistronic, in *C. elegans* they are processed into monocistronic units in a similar fashion as to how *trans*-splicing and polyadenylation are coupled in Trypanosomes[12].

Early global analyses of *trans*-splicing estimated that 70% of *C. elegans* mRNAs are *trans*-spliced[9,13]. A meta-analysis of RNA-seq datasets amended this number to 85% and confirmed a trend that the least expressed genes had a propensity for not being *trans*-spliced[14]. This suggested lack of detection of *trans*-splicing for rare mRNAs rather than a distinct subset of genes bypassing the generic mRNA maturation process. This prompted us to hypothesize that the actual fraction of *trans*-spliced genes could be higher. To test this hypothesis, we decided to apply a different RNA-seq method to improve the characterization of 5′ extremities.

In 2015, Oxford Nanopore Technologies commercialized a new sequencing method relying on the ability of a molecule to affect ionic currents based on the amount of space it takes inside a nanopore, therefore making it possible to reconstruct the sequence of nucleic acids being translocated through a membrane by measuring fluctuations of current[15,16]. In 2020, two teams performed transcriptome-wide analysis using Nanopore direct-RNA sequencing in *C. elegans*. The authors demonstrated that full-length reads can be used for the easy detection of novel splice isoforms. However, the technology does not provide a good enough readout to study *trans*-splicing events[17,18]. A previous study that compared direct-RNA and direct-cDNA sequencing reported that direct-RNA reads exhibit shorter 5′ extremities compared to reads generated with direct-cDNA sequencing[19]. Therefore, we chose direct-cDNA sequencing for amplification-free quantitative analysis of Spliced Leader sequences on the 5′ extremity of *C. elegans* mRNAs. Our experiments reveal how the presence of Spliced Leader sequences affects the Nanopore library preparation process. Our data provide a comprehensive quantitative analysis of SL usage to date in *C. elegans* and allows us to formulate a hypothesis regarding a common feature of non-*trans*-spliced mRNAs.

## Results

### *C. elegans*-spliced leaders interfere with direct-cDNA library preparation

After mapping the reads from three independent direct-cDNA sequencing experiments onto *C. elegans* genome, we verified their

intermediate between the two, and splicing of the SL to the first exon of the pre-mRNA in a second step. The excised 5′ region of the pre-mRNA is called the outron[6].

In 1989, a second SL sequence (SL2) was found that has since been associated with genes located in operon-like clusters[7–9]. Whole-genome microarray analysis has demonstrated a robust correlation between gene clusters and SL2-containing genes[10,11]. In this study, the authors identified more than 1000 clusters for which downstream mRNAs are *trans*-spliced to SL2, yet it remained a mystery how genomic position can affect *trans*-splicing specificity. It was hypothesized

**Table 1 | Minion direct-cDNA runs included in this study**

| Exp name | Primer for 2nd strand synthesis | Basecalled reads | Genome mapping | Transcriptome mapping | | | Strand bias | |
|---|---|---|---|---|---|---|---|---|
| | | | Alignments | Alignments | Genes | Isoforms | Sense | Antisense |
| **SSP_1** | SSP | 1,067,062 | 926,986 | 856,699 | 12,661 | 19,079 | 1.8% | 98.2% |
| **SSP_2** | | 372,188 | 302,164 | 265,555 | 12,586 | 18,351 | 2.4% | 97.6% |
| **SSP_3** | | 428,451 | 269,505 | 215,150 | 11,790 | 17,426 | 6.0% | 93.0% |
| **SSP_4** | | 805,214 | 563,132 | 329,046 | 15,078 | 22,808 | 6.2% | 93.8% |
| **SSP_5** | | 203,384 | 122,662 | 75,874 | 9858 | 13,416 | 3.7% | 96.3% |
| **SSP_6** | | 2,698,484 | 2,003,969 | 520,347 | 12,558 | 19,710 | 6.7% | 92.3% |
| **SL1_1** | SL1 | 7,811,076 | 6,776,420 | 6,015,856 | 15,080 | 24,474 | 26.7% | 73.3% |
| **NP_1** | None | 330,272 | 149,490 | 106,081 | 6736 | 8914 | 1.8% | 98.2% |
| **NP_2** | | 3,238,319 | 2,666,799 | 1,962,645 | 13,041 | 20,486 | 1.7% | 98.3% |
| **NP_3** | | 630,506 | 514,711 | 398,165 | 9781 | 14,107 | 1.4% | 98.6% |
| **NP_4** | | 99,031 | 77,991 | 54,418 | 5343 | 6890 | 1.6% | 98.3% |
| **NP_5** | | 575,203 | 456,843 | 353,154 | 9681 | 14,298 | 1.6% | 98.4% |

*SSP* Strand Switching Primer, *SL1* SL1 primer, *NP* no primer.
Reads from all three conditions were combined for the spliced leader analyses below. More details are provided in Supplementary Tables 1 and 2.

correct alignment in Integrative Genome Viewer (IGV), and noticed an unexpected, reproducible strand bias in favor of antisense reads (Fig. 1a and Supplementary Fig. 1). Out of ~1.3 million reads obtained in these initial experiments, 95% were antisense reads. In addition, we observed that these antisense reads harbored extended 5′ soft-clipped region. The soft-clipped region is the part of the sequence that is beyond the portion of the read that is aligned to the genome. In our case, the soft clips are expected to contain the sequences of oligonucleotide linkers used in the sequencing library preparation as well as the Strand Switching Primer (SSP) and the Spliced Leader sequences on the 5′ end, and polyA tail on the 3′ end. Altogether, it is thus expected to obtain 5′ soft clips of ~80 nt (Supplementary Fig. 2a). By contrast, the 5′ soft-clipped part of the antisense reads had a median size of 416 bp, well beyond the expected size (Supplementary Fig. 2b, c). We investigated the content of these long unaligned sequences. The long soft-clipped from antisense reads, overall, did not contain the expected features, in sharp contrast with the more conventional, but minoritarian sense strand reads (Supplementary Fig. 3a). In addition, we noticed that ~33% of the "long soft-clips" contained a supplementary alignment matching to the sense strand of the original alignment (Supplementary Fig. 3b, c). These supplementary alignments were of lower quality than the primary ones, and we wondered if this could be the result of poor sequence quality in the long soft-clipped region. When plotting the base quality of a single read, we observed the quality was highly variable along the sequence. To mitigate this effect, we plotted the average base quality over a group of ~800,000 individual reads and then computed the mean PHRED-score at every position centered around the 5′ end of the alignment. This confirmed that the base quality of the long soft clips is significantly lower than that of the aligned portion of the reads (Fig. 1b and Supplementary Fig. 4).

The observation of an overwhelming bias for antisense reads and the presence of a low-quality extended 5′ soft clip that could be partially mapped to the cognate sense strand, was indicative of the systematic generation of long hairpin cDNAs. Since this behavior was not reported by other users of the Nanopore direct-cDNA sequencing kit, we hypothesized the origin of this phenomenon might be linked to the presence of *trans*-spliced Spliced Leaders at the 5′ ends of the transcripts. To test this, we selected and aligned full-length reads that displayed a complete SL1 sequence at the beginning of their soft-clipped sequence (Fig. 1c). Looking at those reads, we observed that the first few nucleotides adjacent to the end of the SL1 sequence corresponded to an antisense fragment of itself followed by a portion of the sense cDNA. This reinforced the notion that the two guanine bases at the 5′ extremity of the SL1 sequence had paired with two of the three

cytosines in the middle of the Spliced Leader causing the self-priming of long hairpin cDNAs instead of the expected second-strand synthesis step. The left panel of Fig. 1d illustrates the normal second-strand synthesis step as described in Oxford Nanopore documentation: a Strand Switching Primer (SSP) hybridizes to the terminal cytosines added by the terminal nucleotidyl transferase activity of the reverse transcriptase on the first strand (antisense) cDNA and initiates the synthesis of the sense strand, providing the double-stranded substrate for the adapter ligation that will lead to the generation of an equal proportion of sense and antisense reads during Nanopore sequencing[20,21]. In our *C. elegans* samples the presence of a self-complementary Spliced Leader at the 5′ end of the mRNAs mostly prevents the entry of the SSP and leads to the strand bias we observed. To test this model, we performed additional library preparations replacing the supplied SSP by a SL1 oligonucleotide. In this configuration, the competitive hybridization of the SL1 primer partially prevented the formation of the hairpin and we recovered 27% of sense strand reads (Fig. 1d, right panel) as well as the corresponding fraction of antisense reads with a short soft-clipped region (not shown). We also attempted a direct-cDNA library preparation in the absence of SSP and confirmed it was not required for effective Nanopore sequencing of *C. elegans* transcripts. The obtained strand bias was comparable to that observed in the presence of SSP (see Table 1). Thus, we concluded that the presence of a Spliced Leader sequence (either SL1 or SL2) at the 5′ end of *C. elegans* mRNAs generates double-stranded hairpins of which only the first (antisense) strand can be effectively read by Oxford Nanopore sequencing. Indeed, the helicase currently used is calibrated to provide the constant speed of passage of each individual molecule through the channel by unwinding a double-stranded cDNA molecule. In our case, when the first strand is fully processed the second strand is left single-stranded on the helicase side and likely re-annealing on the other side of the membrane. In this configuration, the speed of passage through the pore becomes too perturbed to provide good quality sequence information for the second strand, thus preventing accurate base-calling for a fraction of SL sequences, and most of the length of the second strand (Fig. 1e).

## Quantitative analysis of spliced leader usage

*C. elegans* spliced leader precursors are expressed from 30 individual *trans*-Spliced Leader Sequence (*sls*) genes: SL1 Spliced Leaders are expressed from 12 genes (*sls-1* to *12*) interspersed in the rRNA cluster located on chromosome V; there are 11 distinct SL2 variants that are produced from 18 *sls-2* genes located on chromosomes I to IV (Fig. 2a). From ~11 million reads collected in three conditions of direct-cDNA

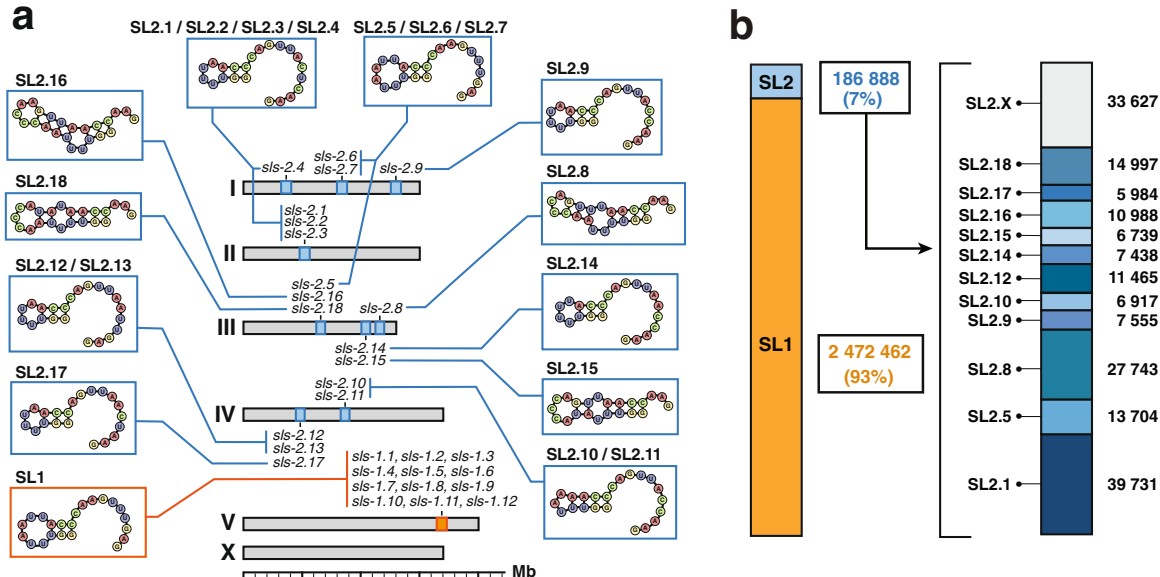

**Fig. 2 | Spliced leaders usage frequency. a** Genomic location of all *C. elegans sls* genes and structure of their 5' hairpin strand. **b** Quantification of reads with high-quality Spliced Leader sequence reveals the usage frequency of each of the *sls* genes. SL2.x indicates reads for which the specific SL2 genes could not be unambiguously identified.

sequencing (see Table 1 and Supplementary Fig. 5), we extracted ~2.8 million reads where the soft-clipped sequence could be unambiguously attributed to a specific Spliced Leader to determine the usage frequency of each SL. The overwhelming majority (93%) of these reads contain the SL1 sequence, as previously reported[13,14,22]. Direct-cDNA sequencing allowed us to measure the relative usage of each SL2 variant. SL2.1, produced by four genes, is the most prominent, representing ~21% of all SL2 reads, followed by SL2.8 (15%) coming from a single gene (Fig. 2b).

For each of the *trans*-splicing sites, we counted the number of reads that could be unambiguously discriminated between SL1 and SL2. For 1008 genes with at least 10 such reads at their most expressed position, we plotted the ratio of SL2/SL1 in relation to the distance to the closest annotated upstream coding gene (Fig. 3a). We confirmed previous observations that *trans*-splicing acceptor sites are strongly favored by one SL sequence or the other and that SL2 *trans*-splicing is strongly preferred when the nearest gene is located ~100 nucleotides upstream of the *trans*-splicing acceptor site[13,14]. However, our data demonstrate that this operon-like genome organization does not fully explain all instances of SL2 proclivity. While 698 out of 1008 genes are located within 200 bp downstream of the closest upstream coding gene, a significant number of SL2 genes are several kb downstream of their closest neighbor. We investigated the possible presence of cryptic termination signals[23] in the upstream region of isolated SL2 genes but did not find evidence supporting this explanation (not shown).

We next measured the proportion of SL2 variants found for each of the preferentially SL2 genes. The distribution showed no evidence of enrichment for any particular SL2 variant supporting a functional redundancy between all of them rather than a sequence-specific recognition mechanism (Fig. 3b).

**Non-*trans*-spliced transcripts display a terminal hairpin that mimics the structure of the spliced leaders**
Our previous meta-analysis of a compendium of RNA-seq data generated by Illumina sequencing indicated a strong positive correlation between the level of expression of a given gene and our capacity to detect a *trans*-splicing event[14]. Here, we confirmed this correlation

between the expression level observed by direct-cDNA sequencing and our capacity to find evidence of *trans*-splicing in either our previous meta-analysis, this study or both (Fig. 4a). This correlation seems to indicate that *trans*-splicing is ubiquitous for *C. elegans* transcripts, which contradicts early observations of non *trans*-spliced mRNAs. We therefore decided to look more quantitatively at the number of reads with direct evidence of *trans*-splicing (*i.e.* reads where the sequence spans the splice site and at least part of the Spliced Leader) relative to the total number of reads for a given gene. For each read alignment, we extracted both start and end positions, corresponding respectively to the 5' and 3' ending position of the cDNA sequence mapped onto the reference genome. For most genes, we observed a dominant start position coinciding with the annotated gene start, which was indicative of full-length reads. For most genes there is a strong positive correlation between the number of full-length reads and the number of reads with detectable SL, which indicates that most genes overwhelmingly produce cDNAs containing a Spliced Leader sequence (Fig. 4b). There is, however, a subset of genes that display significantly less evidence of *trans*-splicing than the majority, and this includes genes that were originally described as "non *trans*-spliced". Those genes, however, still showed the same strand bias as *trans*-spliced genes. Therefore, we investigated their sequences to identify the origin of the underlying hairpins. In all cases, we found that the 5' end of the mRNA had the capacity to generate a hairpin by local self-complementarity (Fig. 4c). For example, the major starting position for a full-length transcript for *vit-2* produces two distinct hairpins that we could identify with seven and six base pairing, respectively.

**Quantitative analysis of spliced leaders usage**
We analyzed the 14,831 genes for which we obtained at least one read that provided a clear sequence of the 5' extremity. Among those, we selected 5647 genes that had at least 20 such reads and found 87.4% are mostly *trans*-spliced (Fig. 3a). Using different threshold for the proportion of trans-spiced reads produces a similar picture of a strong predominance of trans-splicing (Fig. 5b). During this analysis, we observed for a fraction of reads that carried all the hallmarks of hairpin reads (strong antisense bias, long 5' soft clip−see Supplementary Fig. 6) that the sequence of the hairpin structure was not retrieved. We

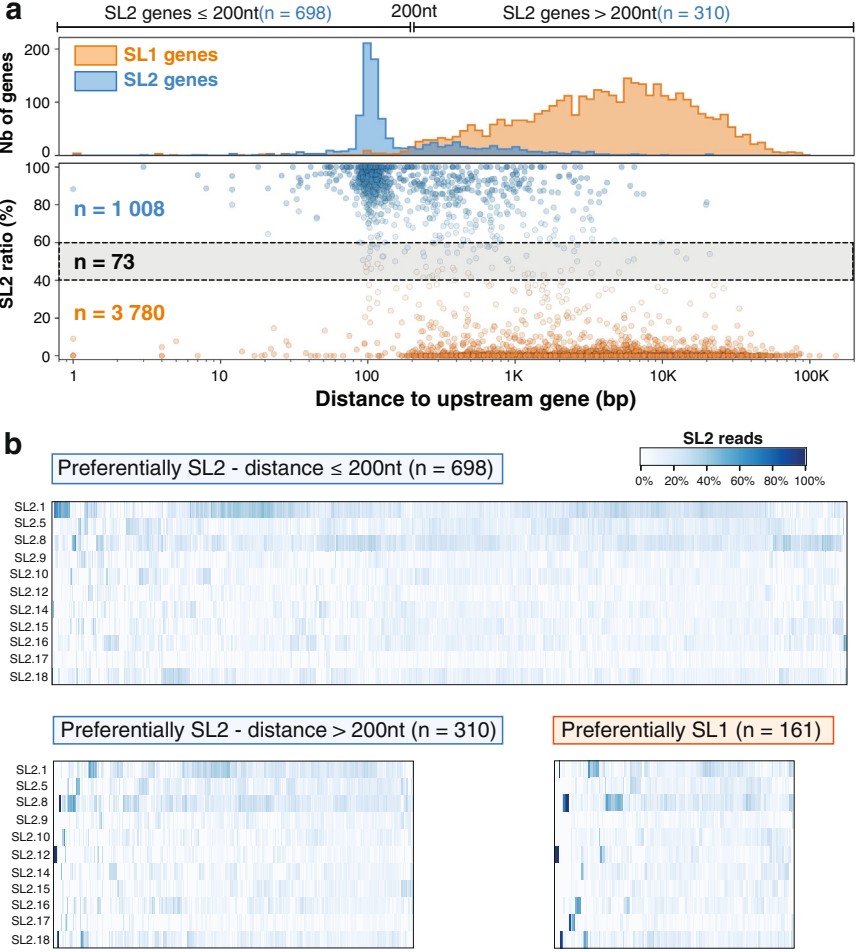

**Fig. 3 | Quantitative analysis of SL sequence selectivity. a** Bottom: A scatterplot of the ratio of SL2/SL1 reads found against the distance to the closest upstream gene. Positions favored by SL2 (SL2 ratio > 60%) are colored in blue and those by SL1 (SL2 ratio <40%) in orange. Top: Distribution of SL1 and SL2 biased positions confirms previous observations of a strong preference for SL2 splicing to occur at a distance of ~100nt downstream from the preceding gene. **b** Each column of the heatmap represents the frequency of usage of each SL2 variant for each SL2 *trans*-spliced site represented in panel **a**. Top: Genes located less than 200nt downstream of another coding gene. Bottom: Genes located further away from the closest upstream coding gene with genes predominantly SL2 on the left and predominantly SL1 on the right.

labeled these "unidentified hairpins reads" as "unidentified" for short in Fig. 5c. We did not find genes with a higher propensity for unidentified reads that could have indicated the absence of any 5' hairpin structure. The 3055 genes that have only "unidentified" reads are characterized by very low expression levels compared to the other categories (Fig. 5c). This is more indicative of a lack of coverage rather than a third mode of mRNA maturation.

We systematically analyzed all start positions for every identified transcript in our direct-cDNA sequencing runs. For each gene", we generated a visual representation of the identified mRNA starts with quantitative information on the usage of each start site and its *trans*-splicing status (see "Data availability"). Figure 6 shows several examples of such representation. For the gene lev-11, which features two promoters with different trans-splicing modes, the distal promoter is the most active but seems to produce almost exclusively mRNAs with endogenous hairpins (89% of reads with sufficient quality to determine the 5'-end structure showed an endogenous hairpin) while the proximal promoter driving the expression of shorter isoforms displays a systematic use of *trans*-splicing. A more expansive representation is the collection of reads obtained for each isoform of a gene along with the different sequences identified within those reads (Supplementary Fig. 7).

## Discussion

In this work, we aimed to provide quantitative information on the 5' ends of *C. elegans* mRNAs through the analysis of full-length sequences. The application of Oxford Nanopore direct-cDNA sequencing strategy to *C. elegans* transcriptome revealed that the presence of Spliced Leaders with 5' self-complementarity causes the self-priming of the cDNA second-strand synthesis during library preparation. While this artifact can be partially circumvented by replacing the regular Strand Switching Primer by a SL1 primer, we did not deem this necessary as the obtained sequences are generally informative up to the end of the antisense strand and a few nucleotides beyond.

Apart from the technical considerations, our observations raise the question of the functional significance of the presence of this stem-loop structure at the extremity of all *C. elegans* SL variants. It has been previously observed that potential stem-loop structures adjacent to the cap are widespread in the Spliced Leaders of phylogenetically divergent species although the sequence of the self-complementarity itself is not conserved[24]. We confirmed the same strand bias in *Leishmania Tarentolae* direct-cDNA sequencing experiments[24], and it is likely that similar library behavior will extend to all species using *trans*-splicing.

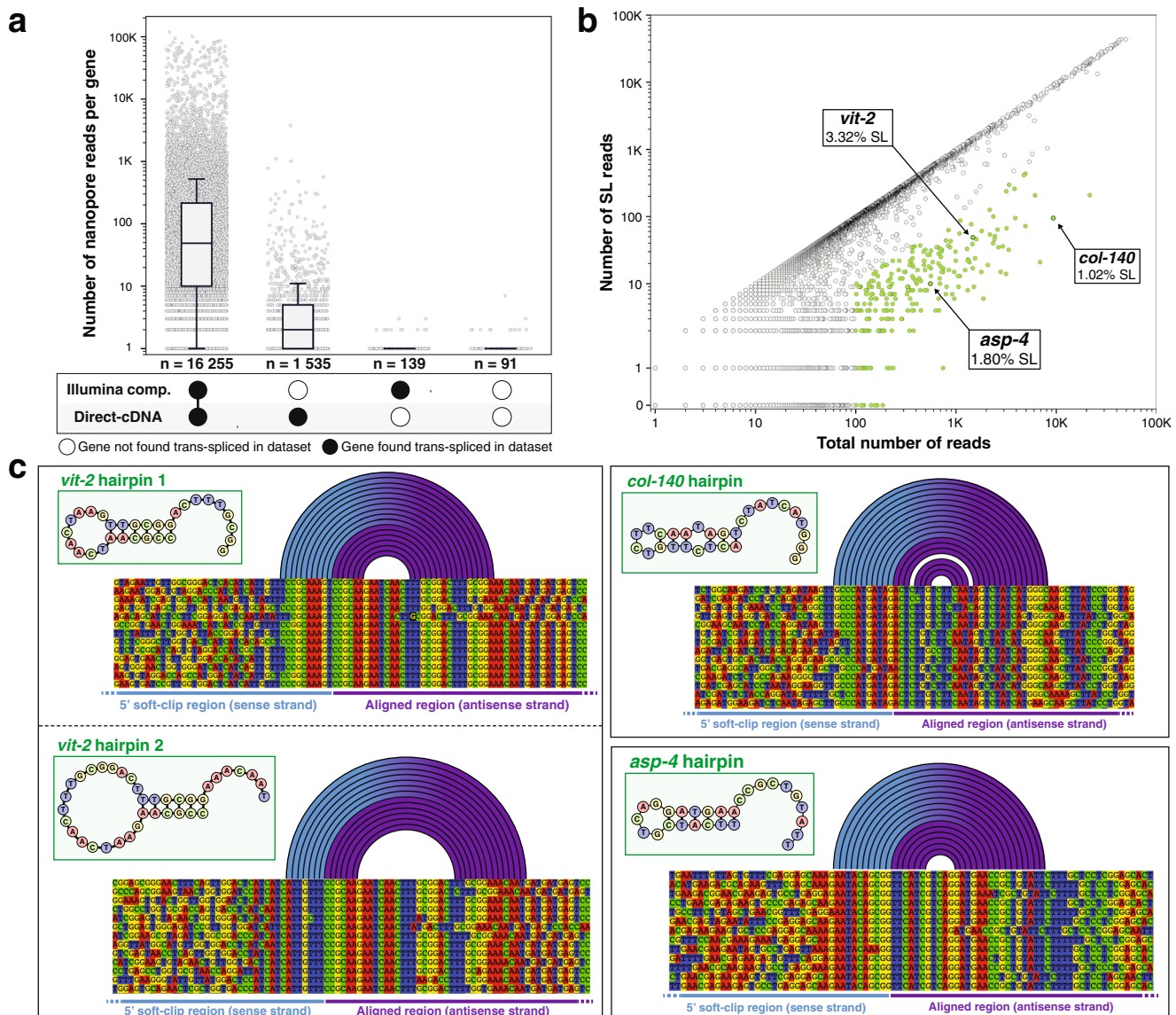

**Fig. 4 | Nature of non-*trans*-spliced mRNAs. a** Gene expression and *trans*-splicing status. We grouped gene in the *C. elegans* genome for which a direct-cDNA read was obtained in this study according to the presence of evidence of *trans*-splicing in this study and/or in the compendium of RNA-seq we previously analyzed[14]. For each gene, we plotted the number of Nanopore reads we obtained as a proxy for gene expression level. The boxes show the quartiles of the dataset while the whiskers extend to show the rest of the distribution, except for points that are determined to be "outliers" using a method that is a function of the interquartile range (python seaborn library). **b** *Trans*-splicing detection level. For each gene, we identified the most frequently detected 5' alignment starting position and plotted the number of reads at this position vs the number of those reads that for which a Spliced Leader sequence could be detected. Genes for which the most represented start position had at least 100 reads but had less than 10% SL detection, are represented in green. **c** Poorly *trans*-spliced mRNAs have the propensity to form a 5' stem-loop structure. For the three genes highlighted in panel b we show a partial alignment of the end of the aligned region and the beginning of the soft-clipped region.

The strong impact we detected on reverse transcribed antisense cDNA indicates a strong propensity for this hairpin formation in vitro which could reflect a similar structure being typical for *trans*-spliced mRNAs. In the cellular context, such a feature could be an important determinant for the binding of proteins involved either in nuclear export or translation initiation. This hypothesis is reinforced by our discovery that non-*trans*-spliced genes in *C. elegans* also harbor a 5' terminal hairpin structure. It has been suggested that translational efficiency of "non-*trans*-spliced *C. elegans* mRNA" is lower than for mRNAs with SLs[25,26]. Our data indicate that these genes are in fact producing a small fraction of *trans*-spliced mRNAs while most of these transcripts bear a terminal hairpin that mimics the structure of SLs (Fig. 7). While it remains possible that some genes produce transcripts devoid of 5' hairpins we did not find any evidence of their existence, the handful of genes that showed minimal strand bias in

the SSP experiments behaved like the majority of hairpin generating genes in other experimental set up (Supplementary Fig. 8). It seems therefore more likely that these genes interacted with the SSP in this set up in a similar manner as trans-spliced genes interacted with the SL1 primer.

It remains to be tested if this reported lower translation rate is due to the suboptimal recruitment of those endogenous hairpins to the translation initiation complex by other protein factors or if their translation is entirely dependent on the small fraction of those mRNAs that have indeed received a Spliced Leader. More detailed studies will be needed as each endogenous hairpin may differently affects both the capacity of its pre-mRNA to be *trans*-spliced and its eventual access to the ribosomes if it is not.

As previously described, we observed SL2 *trans*-splicing to be predominantly targeted to genes located in close downstream

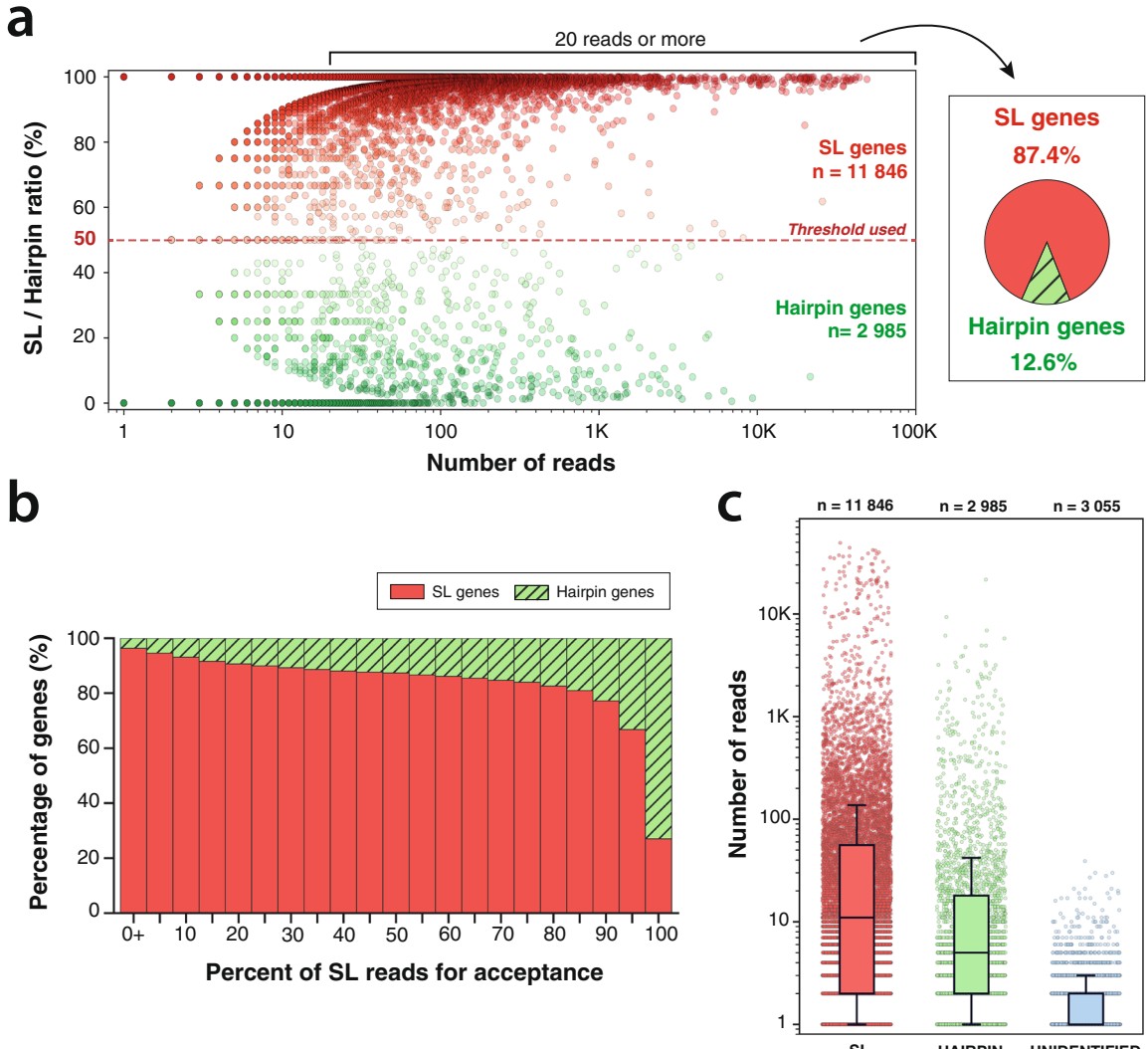

**Fig. 5 | Quantitative analysis of *trans*-splicing prevalence. a** For each gene, we measured the ratio of SL/Hairpin reads at the most expressed starting mRNA position. For genes with at least 20 reads at their most expressed position, we represent the proportion of genes that have a majority of SL reads. **b** Proportion of *trans*-spliced gene with various SL thresholds. **c** Read coverage for genes

majoritarily *trans*-spliced (SL), most with terminal self-complementarity (hairpin) or with unknown 5′ status (unidentified). Boxes show the quartiles of the dataset while the whiskers extend to show the rest of the distribution, except for points that are determined to be "outliers" using a method that is a function of the interquartile range (python seaborn library).

proximity to another gene, supporting a mechanistic link between mRNA 3′ end formation and the downstream addition of SL2 for resolving multicistronic primary transcripts[27,28].

## Methods

### *C. elegans* strains and maintenance
Standard protocols were used for the maintenance of *C. elegans*. Bristol N2 strain was used as wild-type. Nematodes were maintained at 21 °C on Nematode Growth Media (NGM) plates seeded with *E. coli* OP50 as a source of food[29].

### Synchronization of worms
Worms were grown on 10-cm NGM plates to obtain adults gravid worms. We recovered the worms by washing the plates with M9 buffer and then proceeded to perform a bleaching following the established protocol[30]. Upon recovery of the eggs, we let them hatch overnight in M9 buffer deprived of a source of food to trigger a developmental arrest at the L1 stage. The larvae were then put back to growth on 30-cm seeded NGM plates until most of the population was comprised of young adult worms.

### RNA extraction and poly(A) isolations
To remove any bacterial contaminants, synchronized young adults were recovered from their plates with clean $H_2O$ and washed several times. Following centrifugation ($400 \times g$, 1 min), the supernatant was replaced with clean $H_2O$ and the washing step was repeated (at least three times in total). Upon obtention of a clean worm pellet, worms were resuspended in RNA Blue (Top-Bio) reagent and flash froze in liquid nitrogen. We then proceeded to perform a standard phenol-chloroform extraction procedure for RNA isolation. RNA precipitation was performed overnight at 4 °C by adding isopropanol to the lysis supernatant. The pellet was washed in 70% ethanol and dried at room temperature before being resuspended in $H_2O$.

Poly(A) RNAs were isolated from 20 µg of total RNAs using a Dynabeads mRNA Purification Kit from Thermo Fisher Scientific by following the manufacturer's protocol.

### Direct-cDNA library preparation
Sequencing libraries were prepared by following the SQK-DCS109 protocol provided on ONT's website (https://nanoporetech.com)

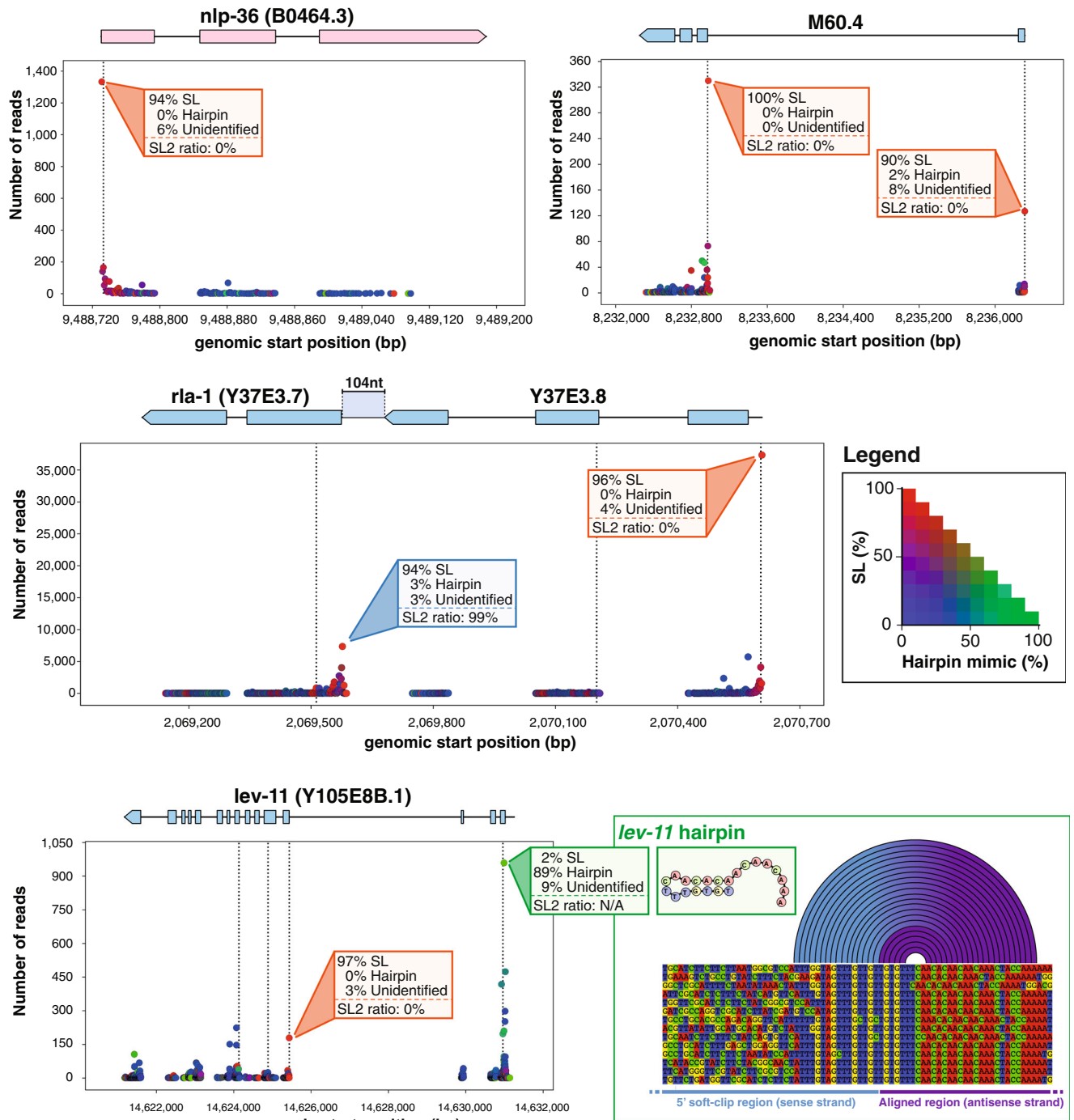

**Fig. 6 | Schematic representation of individual gene *trans*-splicing.** Each alignment start position observed was plotted at the corresponding genomic position with the number of supporting reads. The dots are colored according to the legend on the right, with red indicating a majority of SL reads, green a majority of endogenous hairpin reads and blue reads with no evidence for either. Dotted vertical lines indicate the position of a putative start codon. nlp-

36 displayed a single SL1 *trans*-spliced site. M60.4 presents two *trans*-splicing sites indicative of two alternative promoters with distinct SL1/2 ratios. The Y37E3.7/Y37E3.8 operon with typical SL1 and SL2 preference. *lev-11* gene distal promoter carries an endogenous hairpin while the proximal promoter is mainly SL1.

and by using the reagents provided in their direct-cDNA sequencing kit, along with the recommended 3rd party reagents.

Strand-switching reactions were performed using the Strand Switching Primer provided in the kit (SSP experiments), or by either replacing it with a SL1-specific probe (SL1 experiment) or omitting it from the reaction (no primer experiments) (see Fig. 1d).

**Data acquisition and base-calling**

All sequencing runs were performed using a MinION Mk1B sequencer connected to a computer running the MinKNOW software (version 2.0) with live base-calling disabled. Following library preparation, the sample was directly loaded onto a MinION flow cell (R9.4 chemistry) accordingly to the manufacturer's guidelines. Upon starting a run, we controlled the potency of each flow cell by making sure that a sufficient number of pores was active. The

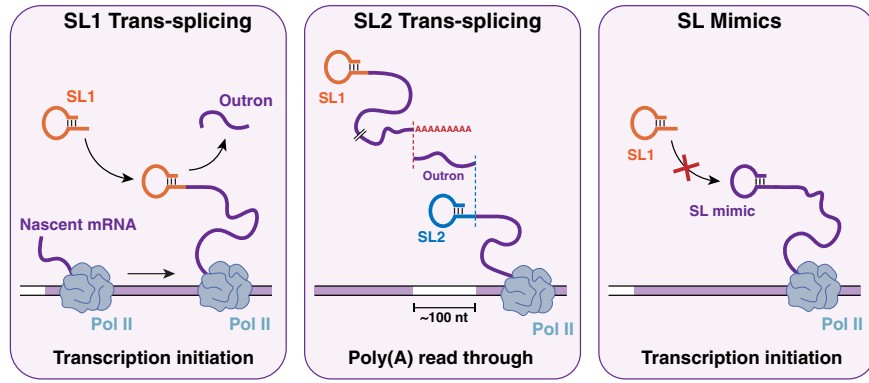

**Fig. 7 | Three types of *C. elegans* mRNA 5' extremities.** mRNAs derived from gene adjacent to their promoters receive mostly SL1. SL2 is preferentially *trans*-spliced to genes located ~100 nt downstream of another gene. Non-*trans*-spliced mRNAs have 5' self-complementarity that mimics the structure of SLs.

experiments were run until we noticed a decrease of activity by monitoring channels' state and general output via the integrated dashboard. Flow cells were then washed and stored for further runs. To avoid cross-contamination, flow cells were only re-used between duplicate experiments. In such cases, we also adjusted the starting voltage due to voltage drift.

Base-calling of the obtained raw data (fast5 format) into readable files (fasta format) was performed using Guppy (version 2.3.7) with the flip-flop algorithm for better accuracy (options: c dna_r9.4.1_450bps_fast.cfg trim_strategy 0 qscore_filtering 1 calib_detect 1). Additionally, the base-calling step was performed on virtual machines running on Google cloud Platform and associated with an NVIDIA Tesla V100 GPU to enable guppy's GPU base-calling (options: x auto num_callers 16 chunks_per_runner 768 -chunk_size 500 gpu_runners_per_devices 8).

### Read alignment and processing
Following base-calling, reads were mapped with minimap2 v2.16 against *C. elegans* reference genome and transcriptome (WBcel235/ce11 genome assembly) obtained from the wormbase release WS270 (https://wormbase.org/).

Genomic alignments were generated by mapping the obtained cDNA sequences against *C. elegans* chromosomal sequences using the splice-aware alignment preset to account for the presence of intronic sequences in the genome (command: minimap2 ax splice). Furthermore, transcriptomic alignments were generated by using the mRNAs sequences as the reference along with minimap2 preset for mapping Oxford nanopore reads (command: minimap2 ax mapont).

The resulting alignment files, obtained in SAM format, were then sorted, indexed, and converted to BAM format using SAMtools[31].

### Genome browser visualization
Genomic alignments were visualized using the Integrated Genome Viewer (IGV)[31,32].

### Measure of strand bias
The measure of strand bias in each experiment was performed by using a custom python script which parsed all transcriptomic alignments and detected whether the read had been reversed during the alignment step or not. Since the reference sequences are in the 5' to 3' orientation, independently from the gene orientation, it is therefore possible to detect the strand origin of any given read by checking if the read had been reversed during mapping (antisense strand) or not (sense strand).

### Base-quality assessment
When extracting base quality from a single Nanopore read, we observed a lot of intrinsic variability. Therefore, to compare the quality of the 5' soft-clipped region (unaligned) with that of the aligned

portion of the reads, we used the following method: (1) we randomly selected many reads; (2) extracted their base quality; (3) determined the position of each base, relative to the alignment's start; (4) computed the mean base quality for each position; (5) plotted the average base quality obtained. An illustration of the method is shown in Supplementary Fig. 9.

### Identification of splice leader sequences
Due to the nature of *trans*-splicing, we reasoned that a full-length cDNA read would map onto its transcript reference from the *trans*-splicing site to the polyadenylation site, leaving an unmapped (soft-clipped) region on each end, one in 5' (the splice leader sequence) and one in 3' (the poly(A) sequence). Hence, we decided to search for the splice leader sequence directly upstream of the alignment start. Moreover, to account for the inherent noise of Nanopore reads and the small size of the SL motif (~23nt), we decided to add some tolerance to allow for the detection of closely related sequences.

The method is briefly described as follows: (1) we parsed transcriptomic alignments and extracted the sequence (up to 100 nt) directly upstream the alignment start; (2) for each annotated SL sequence, we performed a semi-global alignment onto the extracted sequence; (3) the alignment score was then evaluated using a custom threshold; (4) if the score is equal or higher than the threshold, the SL is considered detected and we start evaluating another SL sequence, otherwise the sequence is shortened (2 nt in 5' are trimmed from the sequence) and we repeat the previous steps until we reach a truncated SL sequence of 7 nt, in which case the SL is considered undetected; (4) After evaluating every SL sequence, we then retrieved the SL sequence which scored the best overall. Reads for which we detected both a SL1 and a SL2 sequence are considered SL.X and reads for which we detected several SL2 sequences are considered SL2.X.

### Measure of splice leader usage
To produce a statistically relevant analysis, we decided to measure SL ratio solely based on confident SL matches. Therefore, for any gene analyzed, we considered exclusively matches with an alignment score superior to 9 or in the direct vicinity of their alignment start (distance of 2 nt or less). Doing so allowed us to include any match resulting from the detection of poorly sequenced SL motifs (due to a drop of base quality as seen in Fig. 1b) but for which our confidence in the result is strengthen by their presence near the alignment start, where splice leader sequences are expected to be found. This is supported by the fact that high-scoring matches (>15) are preferentially found just upstream the alignment start, whereas low-scoring (<10) can be found all across the 5' soft-clip sequence, but also with a noticeably higher frequency directly upstream the alignment start (Supplementary Fig. 10).

Confident SL matches were then used to measure the ratio of SL2 to SL1 reads for a given gene or a given position.

NB: In order to avoid false positive SL1 counts potentially caused by heterologous binding of SL1 primer during the second-strand synthesis, we only considered antisense reads (first strand cDNAs) from the SL1 experiment.

### Identification of 5′ endogenous hairpins

To identify the presence of a stem-loop we examined the complementarity between two regions of the same read using the following method: (1) we selected reads originating from the antisense strand of the cDNA and for which no SL had been detected; (2) we extracted the 5′ unaligned region corresponding to the potential complementary stem (positions −13 to +2 relative to alignment start); (3) the 5′ unaligned region extracted was then reverse complemented and mapped against the aligned region; (4) if at least 12 nt (out of 15) were successfully mapped, we validated the presence of an endogenous hairpin in the read.

### Visualization of secondary RNA structures and hairpin motifs

We used the RNAfold web server (http://rna.tbi.univie.ac.at/cgi-bin/RNAWebSuite/RNAfold.cgi) to generate secondary RNA structures based on SL or 5′ endogenous hairpins sequences. Additionally, we use the RNAbows web server (http://rna.williams.edu/rnabows) to compute the probability of base pairing between regions of the same read.

### Reporting summary

Further information on research design is available in the Nature Portfolio Reporting Summary linked to this article.

## Data availability

The direct-cDNA datasets used in this study are listed in Supplementary Table 2 and are available in the Sequence Read Archive (SRA) under the accession code PRJNA822363. The dataset table and all other files generated in the current study are available at https://figshare.com/search?q=DOI%3A+10.6084%2Fm9.figshare.19131260. A companion web-app, allowing the reader to view any gene found in our sequencing experiments, can be found at https://elegans-trans-splicing.streamlit.app.

## Code availability

The scripts and methods used in this study are available via https://github.com/florianbrnrd/elegans-trans-splicing.

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

## Acknowledgements

F.B. was funded by IDEX Bordeaux International Doctorate Program, ISF grant 1339/17, and Sandwich scholarship from the Council of Higher Education. O.R. is grateful to funding from the Eric and Wendy Schmidt Fund for Strategic Innovation (Polymath Award #0140001000) and funding from the Khan foundation.

## Author contributions

Worm culture and RNA sequencing were performed by F.B and D.Da. under the supervision of D.Du. and O.R. Data analysis was performed by F.B. and D.Du. Python scripts and figures were produced by F.B. under the supervision of D.Du. D.Du. and F.B. wrote and edited the manuscript.

## Competing interests

The authors declare no competing interests.
