## [Peer Review File · Nature Communications]

Quantitative analysis of *C. elegans* transcripts by Nanopore direct cDNA sequencing reveals terminal hairpins in non spliced mRNAsREVIEWER COMMENTS

Reviewer #1 (Remarks to the Author):

In this manuscript by Bernard et al., direct (PCR-free) cDNA sequencing using the Oxford Nanopore Technologies (ONT) platform was leveraged to overcome technical challenges inherent to short read sequencing technologies and ONT direct RNA sequencing in comprehensively profiling *C. elegans* trans-splicing. Unexpectedly, the authors discovered a strand bias from ONT DNA sequencing in which most of the sequencing reads were antisense in orientation due to mRNA hairpin formation that impeded second strand synthesis during library preparation and interfered with pore chemistry (Figure 1). The majority of these antisense reads comprised trans-spliced transcripts containing the SL1 spliced leader sequence (Figure 2). Another subset of antisense reads contained an SL2 spliced leader sequence, and the authors characterized the usage frequency of 11 distinct SL2 variants (Figure 3). In addition to these trans-spliced transcripts with strong antisense strand bias, the authors identified a third subset of antisense reads that lacked a spliced leader sequence. Further sequence analysis demonstrated that these non-trans-spliced transcripts still displayed antisense strand bias due to hairpin formation at the 5' end based on local self-complementarity (Figure 4). Base quality comparisons of reads with either a spliced leader sequence or endogenous hairpin or neither revealed that the latter fraction of reads had lower base quality scores, precluding complete mapping at the 5' end and hairpin identification (Figure 5). However, for full-length reads, spliced leader sequences were identified among 75% of reads with various SL1/SL2 dynamics (Figure 5). The cDNA sequencing datasets used in this manuscript are accessible through the Sequence Read Archive, and scripts to perform the analysis are available on a GitHub repository.

Overall, the work in this manuscript corroborates much of what has been previously published about *C. elegans* trans-splicing in the literature, but also provides new insight into SL2 variant usage, SL1/SL2 trans-splicing dynamics, and endogenous hairpin formation of transcripts lacking a spliced leader sequence. Of note, the authors report an interesting artifact in ONT DNA sequencing due to transcript hairpin formation that has not yet been reported and is important for studies in other systems where trans-splicing and/or endogenous hairpin formation is prevalent. This manuscript should be accepted with minor revisions, the most critical being providing additional quality control metrics of the sequencing data to ensure that analysis was rigorous.

Major criticisms:

1. Quality control:

- a. It is unclear if any sort of read filtering (e.g., base quality, overall poor alignment, large insertions, large 3' soft-clips, reads without poly-A tail based on QC tag) was performed to only retain high-quality reads for analysis. This is particularly important to accurately characterize trans-splicing and endogenous hairpin prevalence, and Figure 5 specifically makes comparisons between "all reads" and "full-length reads".
- b. How were full-length reads defined/identified and did they reflect the structure and length of transcripts in the reference transcriptome?
- c. What were the mean and median read lengths for each of the replicates in the three datasets? Were they comparable?

2. It is unclear how and why 5' soft-clips were classified as either short or long. The Results section simply discusses the "long soft-clipped region". Supplementary Figure 1 only states that "in our experiments, 5' soft-clip regions are unexpectedly long". With such a vague definition, the importance of the difference in 5' soft-clip region length is lost and makes interpretation of Supplementary Figure 2 difficult.

3. Figure 5c nicely presents some examples of genes with various SL1/SL2 trans-splicing dynamics, and the companion web-app provides results for all genes; however, this makes it difficult to

appreciate trans-splicing dynamics on a comprehensive basis. Would it be possible to further quantify the different categories of trans-splicing dynamics highlighted in this figure? For example, for how many genes is the trans-splicing dynamics strictly restricted to one of the splice leader sequences? And if restricted to SL2, is it restricted to a specific SL2 variant?

4. Figure 6: This model figure makes it appear as though there are only either trans-spliced transcripts or non trans-spliced transcripts with endogenous hairpins, but Figure 5b showed that there is a significant subset of transcripts that are not trans-spliced or have endogenous hairpins. Also, it may be helpful to add the percentage of transcripts with each of the three types of 5' extremities so that readers can appreciate the findings on a more transcriptome-wide scale.

Minor criticisms:

1. Abstract: The first sentence states that "nematode mRNA processing involves a trans-splicing step through which a 21 nt sequence ...". I do not have extensive knowledge about trans-splicing and may be mistaken, but I believe that it is actually a 22 nt sequence, not 21 nt.

2. Figure 1b: The figure legend states that "~800,000 individual Nanopore reads were measured", but the second-to-last paragraph in under C. elegans Spliced Leaders interfere with direct-cDNA library preparation states that "we plotted the average base-quality of groups of one million reads". Furthermore, it may be helpful to readers if the same term, either PHRED score or Q-score, was used in the text and corresponding figure for consistency.

3. Supplementary Figure 2: The barplots and figure legends do not appear to match. Barplots displayed are for sense reads and reads with short and long 5' soft clips. The title of this figure only refers to short soft clips, and the legend mentions 3' soft clip regions for which there is not a barplot. Moreover, the difference between short and long 5' soft clips and its significance is not clear since this has not been clearly described in the manuscript. It may also be unclear to those not familiar with the ONT cDNA sequencing library preparation protocol that the abbreviation SSP in the legend stands for Strand Switching Primer.

4. Supplementary Figure 3: Were short and long 5' soft-clips combined to generate the top histogram? If so, should they perhaps be plotted separately since long 5' soft-clips are described as being "unexpectedly long" in the legend in Supplementary Figure 1?

5. There are some paragraphs and figures in the manuscript that would benefit from additional clarification:

a. Under C. elegans Spliced Leaders interference with direct-cDNA library preparation:

i. It is unclear which datasets were ultimately analyzed. The third sentence mentions "~1,3 million reads obtained in these initial experiments" whereas under A significant fraction of SL2 trans-spliced genes are not part of an operon, the fourth sentence mentions "~11 million reads collected in our direct cDNA sequencing". It is unclear if "initial experiments" refer to those generated using only SPP and if all three types of datasets (SPP, SL, and NP) were combined in latter analyses.

ii. Sentence 6 states that soft-clips are expected to contain certain elements, and so it is a little confusing to read in the next sentence that "indeed, that was the case for the minority of sense strand reads" without any further clarification for this unexpected observation. I assume only a minority of sense strand reads contained all expected elements because hairpin formation prevented complete second strand synthesis. If so, it may be helpful to the reader to allude to this.

b. Figure 5b: What percentage of full-length transcripts were identified from all reads?

Reviewer #2 (Remarks to the Author):

Many medically, ecologically, and economically important groups of organisms utilise spliced leader *trans*-splicing to replace the nascent 5' untranslated regions of their transcripts. This process also allows these organisms to organise their genes into concerted transcription units, eukaryotic operons, with the spliced leader providing a cap structure to otherwise uncapped transcripts. Given that both spliced leader *trans*-splicing and operon usage is found in many eukaryotic groups, the impact of both processes on the eukaryotic transcriptome is an important topic for investigation.

Other than the splice donor site and the Sm binding site, there is no sequence conservation of spliced leader RNAs between higher order phylogenetic groups, but the formation of a hairpin within the spliced leader appears to be a highly conserved feature. The authors of this study show convincingly that the presence of this hairpin motif interferes with the strand-switching primer based polymerase used to construct Oxford Nanopore Technologies (ONT) direct-cDNA sequencing system libraries. This leads to an extreme bias towards antisense reads and a preponderance of 5' soft-clipped, error-prone tails corresponding to the sense strands of full length transcripts.

Given that long read sequencing technology is an important tool in the characterisation of transcriptomes, the work is important to all *C. elegans* researchers using or contemplating using ONT transcriptome sequencing. However, since most, if not all, spliced leaders have this same propensity to form 5' hairpins, it is likely to be an issue for all organisms whose transcripts are substantially spliced leader *trans*-spliced. In fact, the authors could strengthen the manuscript by explicitly pointing this out.

The work is clearly and beautifully presented, including a convincing mechanistic explanation for the poor sequencing quality of the sense-strand soft-clipped 5' tails. Moreover, I can independently confirm that this is a reproducible issue associated with this technology. My laboratory's own data generated using the same direct-cDNA sequencing system contains a similar degree of read bias and extended soft-clipped reads. It is gratifying to have an explanation for this issue.

One thing here, though, is that the authors are not showing any alignment statistics; it's not clear if the alignments could be improved. There are two things that could help with the presentation of this issue. 1) Have the authors tried to trim and orient the reads using pychopper? This would have directly alerted them to the excess of VNP-VNP reads and allowed them to quantify the problem directly before alignment. 2) The reads could also have been screened for spliced leader 5' tails (and those tails then removed) before alignment.

Despite these important technical issues, the depth of sequence coverage allows the authors to go on to comprehensively study *C. elegans* spliced leader usage. To some degree, this aspect of the work does not appreciably add to our current understanding of spliced leader *trans*-splicing in *C. elegans*, but as the authors state, it is the most comprehensive to date.

However, I am puzzled by the title: "A significant fraction of SL2 *trans*-spliced genes are not part of an operon". From what is presented, the authors claim this on the basis that there are many (approximately 300) SL2 *trans*-splicing events greater 200 bp (and up to several kb) downstream of the closest upstream gene. But we know that such distances do not rule out the possibility of being in an operon, as outlined in Morton and Blumenthal, 2011 (Morton JJ, Blumenthal T. Identification of transcription start sites of *trans*-spliced genes: uncovering unusual operon arrangements. *RNA*. 2011;17: 327-337. See also, Blumenthal T, Davis P, Garrido-Lecca A. Operon and non-operon gene clusters in the *C. elegans* genome. *WormBook*. 2015; 1-20.). I would

recommend that the section is re-written to consider this information and to change the sub-heading. The evidence as presented does not support the sub-title's assertion.

The most intriguing outcome of this analysis is the identification of non-*trans*-spliced transcripts that nonetheless possess inherent 5'-hairpins. This is striking and suggests the possibility that cellular adaptation to the preponderance of transcripts with 5' spliced leaders *C. elegans* has created a selective pressure that has shaped non-*trans*-spliced transcripts.

Finally, there are a couple of points raised by the Discussion that I would like to address. As noted above, the presence of hairpins in spliced leaders from other nematodes, animals and protists means that the issues documented here will apply more broadly. This should be noted and expanded upon in the discussion since it will make the impact of this work more explicit.

There is also a statement that requires clarification: "Additionally, while all Spliced Leaders present a modified Guanine in 5', endogenous terminal hairpins do not, which would make them uncapped mRNAs". The authors appear to be stating that the non-spliced leader *trans*-spliced transcripts lack a cap – this is not (and cannot be) the case, they will have a standard monomethyl guanosine cap. I presume that the authors simply mean that they will not have trimethyl guanosine caps.

Minor comments and typographical errors

The prefixes in *trans*/*cis*-splicing should be italicised.

Abstract 1st line, the text states "21 nt sequence" – nematode spliced leaders are 22 nt

"While 700 out of 1011 genes are located within 200 bpb downstream..."

The authors use the term "messengers" at several points in the manuscript – this term is ambiguous; "mRNA" or "transcripts" would be better.

There are quite grammatical infelicities/errors throughout the manuscript. I detail a few below, but the authors should check through the revised manuscript carefully.

"...resulting long RNAs needs to be matured..."

"In this protists, genes do not contains introns, therefore the spliceosome's sole function..."

Jonathan Pettitt

Reviewer #3 (Remarks to the Author):

[Bernard et al.,] describe an interesting set of observations resulting from the application of cDNA-based long read sequencing with the oxford nanopore device. Their study suggests that RNA hairpins present in *C. elegans*' spliced leaders (SLs) allow for self-priming during cDNA synthesis. They go on to show several examples of non-SL genes that exhibit similar hairpin structures, suggesting that this may be a widespread feature of mRNAs in the worm.

The observations are interesting and novel. I have a handful of concerns that, if remedied, would

significantly strengthen the study.

Their interpretation of the species in question relies on a novel technology

1. The key step for their sense-antisense reads seems to be simple self-priming and reverse transcriptase extension. I would feel more confident in their interpretation of the reads if they could show the species existed with canonical methods. For example, RT, PCR, and sanger sequencing across a few of the most abundant hairpins is straightforward and would strengthen their conclusion as to the origin and nature of the reads. This is key as the nanopore technology is still relatively new, and thus certainly has unanticipated misinterpretations.

2. The authors conclude that many SL2s are not part of operons. The key data is omitted: "We investigated...cryptic termination...but didn't find evidence (data not shown)." It is still possible the SL2s are part of operons, but the upstream RNA/polyadenylation events from whence they are derived are unstable or rare. The data presented does not speak to this possibility, and ultimately does not support their conclusion.

3. The authors suggest that many non-SL mRNAs have a terminal hairpin. The data presented is anecdotal, showing four examples (Fig 4C). It's clear they have many more non-SL mRNAs (Fig 4B). It would be helpful (and more rigorous) to have a statement about the generality of the hairpin phenomenon for the mRNAs in Fig 4B. How many of non-SL mRNAs have a recognizable hairpin? (Perhaps this is what is deemed "unidentified"/"unknown" in Fig5B, but this is not clear.)

4. I am apprehensive about trusting RNA folding algorithms, which will readily produce hairpins on random sequence. The nature of the hairpin and the sense/antisense artefact is well-suited to a more rigorous test of the terminal hairpin forming behavior of mRNAs, SL or not. By examining the frequency of sense/antisense reads at a given locus, one should be able to get an accurate estimation of the terminal hairpin propensity. For mRNAs with SLs, the sense/antisense ratio should be ~5%/95%, and for mRNAs without terminal hairpins it should be ~50%/50% (Fig 1D). This would be a rigorous way of testing the hairpin propensity of the non-SL transcripts in general.

5. The authors suggest that the proportion of trans-spliced mRNAs in elegans is higher than earlier estimates of 70% or even 85%. However, the authors provide no number. It would be informative for the authors to share their estimate of the fraction trans-spliced mRNAs.

6. The authors suggest that non-SL mRNAs are uncapped: "endogenous terminal hairpins do not, which would make them uncapped mRNAs." My understanding is that this is inaccurate. Indeed, transcription start site mapping by others (PMID: 23260138, 23636945) has used the capped nature of non-SL mRNAs to map their 5'ends. The authors should correct this statement.

7. The axes in Fig 4A/B seem confusing and perhaps show the wrong scale. They indicate some genes have tens of thousands of nanopore reads, with some nearing 100,000. The authors libraries only have a few hundred thousand reads each, so it would seem unlikely that even a highly abundant (~1% of mRNA) would achieve such high counts. Is the axis mislabeled? Were libraries combined?

8. It would be beneficial if the authors took some care to ensure typos and ambiguities in communication are removed. While I could read through most of the small typos, others left me wondering what the authors were attempting to say. For example, "...for a large number of reads the sequence of the hairpin structure was not accessible." What does "accessible" mean in this context? Does this mean "We were unable to identify the sequence of the hairpin structure."? Does that mean there was no hairpin structure, or that the precise position at which antisense became sense was ambiguous? I am not sure, was not able to figure it out from the rest of the section, and thus not sure what is being displayed in Figure 5B.

Please find below our response to the reviews for our Manuscript NCOMMS-22-14730
Original reviewers comments are in red
When our response led to a modification of the manuscript text we included the text in purple

REVIEWER COMMENTS

Reviewer #1 (Remarks to the Author):

In this manuscript by Bernard et al., direct (PCR-free) cDNA sequencing using the Oxford Nanopore Technologies (ONT) platform was leveraged to overcome technical challenges inherent to short read sequencing technologies and ONT direct RNA sequencing in comprehensively profiling *C. elegans* trans-splicing. Unexpectedly, the authors discovered a strand bias from ONT DNA sequencing in which most of the sequencing reads were antisense in orientation due to mRNA hairpin formation that impeded second strand synthesis during library preparation and interfered with pore chemistry (Figure 1). The majority of these antisense reads comprised trans-spliced transcripts containing the SL1 spliced leader sequence (Figure 2). Another subset of antisense reads contained an SL2 spliced leader sequence, and the authors characterized the usage frequency of 11 distinct SL2 variants (Figure 3). In addition to these trans-spliced transcripts with strong antisense strand bias, the authors identified a

third subset of antisense reads that lacked a spliced leader sequence. Further sequence analysis demonstrated that these non-trans-spliced transcripts still displayed antisense strand bias due to hairpin formation at the 5' end based on local self-complementarity (Figure 4). Base quality comparisons of reads with either a spliced leader sequence or endogenous hairpin or neither revealed that the latter fraction of reads had lower base quality scores, precluding complete mapping at the 5' end and hairpin identification (Figure 5). However, for full-length reads, spliced leader sequences were identified among 75% of reads with various SL1/SL2 dynamics (Figure 5). The cDNA sequencing datasets used in this manuscript are accessible through the Sequence Read Archive, and scripts to perform the analysis are available on a GitHub repository.

Overall, the work in this manuscript corroborates much of what has been previously published about *C. elegans* trans-splicing in the literature, but also provides new insight into SL2 variant usage, SL1/SL2 trans-splicing dynamics, and endogenous hairpin formation of transcripts lacking a spliced leader sequence. Of note, the authors report an interesting artifact in ONT DNA sequencing due to transcript hairpin formation that has not yet been reported and is important for studies in other systems where trans-splicing and/or endogenous hairpin formation is prevalent. **This manuscript should be accepted with minor revisions**, the most critical being providing **additional quality control metrics of the sequencing data to ensure that analysis was rigorous**.

Major criticisms:

1. Quality control:

- a. It is unclear if any sort of read filtering (e.g., base quality, overall poor alignment, large insertions, large 3' soft-clips, reads without poly-A tail based on QC tag) was performed to only retain high-quality reads for analysis. This is particularly important to accurately characterize trans-splicing and endogenous hairpin prevalence, and Figure 5 specifically makes comparisons between "all reads" and "full-length reads".

Early on in our analysis we realized that about 20 to 30% of "failed" reads could be **successfully** mapped to the transcriptome. So as to not discard these reads, we decided to include in our analysis all the reads that could be mapped to a specific transcript. We added **Supplementary Table 2** that indicates the status of pass/fail of the reads and the features that could be found within them.

- b. How were full-length reads defined/identified and did they reflect the structure and length of transcripts in the reference transcriptome?

The term full-length in the original manuscript was ambiguous. It is frequent to find reads that include SL sequence even if they seem to come from an internal (potentially spurious) start site. In this case the read is a full-length in the sense that it captures the complete sequence of a mRNA molecule but it may not cover the entirety of a predicted isoform. Technically, all our reads containing either SL or hairpin sequences are “Full-Length” reads, in that we were able to read the entirety of the mRNA molecule from 3’ to 5’. However, our data highlights the fact that spurious transcription initiation contributes a significant amount of messengers that are detected by RNA-seq experiments and also display a Spliced Leader.

To fully take advantage of the quantitative nature of our data, we decided to focus on the most detected alignment start position for each gene to define the “main isoform” for each gene. To avoid confusion we elected to not use the term “full-length” in the revised manuscript.

c. What were the mean and median read lengths for each of the replicates in the three datasets? Were they comparable?

We generated a figure displaying the distributions of read lengths and alignment lengths obtained in each sequencing run. The outputs are very similar between all experiments with the exception of the SL1-primed second strand run which contains a higher proportion of shorter reads consistent with a higher proportion of non-hairpin cDNAs (**Sup. Fig 3**).

2. It is unclear how and why **5’ soft-clips were classified as either short or long**. The Results section simply discusses the “long soft-clipped region”. Supplementary Figure 1 only states that “in our experiments, 5’ soft-clip regions are unexpectedly long”. With such a vague definition, the importance of the difference in 5’ soft-clip region length is lost and makes interpretation of Supplementary Figure 2 difficult.

We grouped and modified the original Sup. Fig 1 and 3 into the new **Sup. Fig 1** to clarify this point and better explain the notion of short and long 5’ soft-clips.

3. Figure 5c nicely presents some examples of genes with various SL1/SL2 trans-splicing dynamics, and the companion web-app provides results for all genes; however, this makes it difficult to appreciate trans-splicing dynamics on a comprehensive basis. Would it be possible to further quantify the different categories of trans-splicing dynamics highlighted in this figure?

For example, for how many genes is the trans-splicing dynamics strictly restricted to one of the splice leader sequences? And if restricted to SL2, is it restricted to a specific SL2 variant?

We explored the preferential usage of Spliced Leaders in Figure 3.

We modified Figure 3a to make it more easy to see the number of genes with mostly SL1 or SL2 *trans*-splicing according to their relative usage. Figure 3b shows that SL2 genes do not display significant preference for any particular SL2 variant.

4. Figure 6: This model figure makes it appear as though there are only either trans-spliced transcripts or non trans-spliced transcripts with endogenous hairpins, but Figure 5b showed that there is a significant subset of transcripts that are not *trans*-spliced or have endogenous hairpins.

In the original manuscript, the fraction of reads that were “unidentified” corresponded to reads that have evidence of being hairpin (with long 5’ soft-clip) but for which we were unable to pinpoint the nature of the causal hairpin. Those reads still bear the hallmarks of hairpin cDNA reads but the sequencing results did not allow us to identify the origin of the hairpin that caused the second strand synthesis.

We collated the informations relative to those “unidentified hairpins” reads in a new visual representation that has been added to our online visualization tools and is presented in the (see **Sup. Figures 4 and 5**)

Also, it may be helpful to add the percentage of transcripts with each of the three types of 5' extremities so that readers can appreciate the findings on a more transcriptome-wide scale.

We have clarified this point :

-The new **Sup. Figure 2a** breaks down the quantification of library linkers in soft-clipped sequences for the different read types.

-The new **Sup. Figure 6** shows the features of every reads for each isoform of each gene. As suggested by the reviewers, this allows us to better understand how was obtained the categorization of each gene according for their preference. It also provides a more expansive representation of all the data collected than the summary we present in Figure 6.

Minor criticisms:

1. **Abstract:** The first sentence states that “nematode mRNA processing involves a trans-splicing step through which a 21 nt sequence ...”. I do not have extensive knowledge about trans-splicing and may be mistaken, but I believe that it is actually a 22 nt sequence, not 21 nt.

Indeed, SL1 adds 22nt but some variants of SL2 add only 21nt. In order to introduce the concept for nematodes and trypanosomes at large which includes various different specific Spliced Leaders sizes we removed the inaccurate number in favor of a more general statement:

“In nematode and kinetoplastids, mRNA processing involves a *trans*-splicing step through which a short sequence from a snRNP replaces the original 5' end of the primary transcript.”

2. **Figure 1b:** The figure legend states that “~800,000 individual Nanopore reads were measured”, but the second-to-last paragraph in under C. elegans Spliced Leaders interfere with direct-cDNA library preparation states that “we plotted the average base-quality of groups of one million reads”. Furthermore, it may be helpful to readers if the same term, either PHRED score or Q-score, was used in the text and corresponding figure for consistency.

We thank the reviewer for noticing this inconsistency. We fixed text to match the figure legend in the revised manuscript.

“To mitigate this effect, we plotted the average base-quality of groups of ~800,000 reads and then computed the mean PHRED-score at every position centered around the 5' end of the alignment.”

3. **Supplementary Figure 2:** The barplots and figure legends do not appear to match. Barplots displayed are for sense reads and reads with short and long 5' soft clips. The title of this figure only refers to short soft clips, and the legend mentions 3' soft clip regions for which there are is not a barplot. Moreover, the difference between short and long 5' soft clips and its significance is not clear since this has not been clearly described in the manuscript. It may also be unclear to those not familiar with the ONT cDNA sequencing library preparation protocol that the abbreviation SSP in the legend stands for Strand Switching Primer.

The Supplementary Figure 2 clarifies the definition of short and long 5' soft-clips. We fixed the legend of Sup. Fig 2 according to this reviewer's recommendations.

4. Supplementary Figure 3: Were short and long 5' soft-clips combined to generate the top histogram? If so, should they perhaps be plotted separately since long 5' soft-clips are described as being "unexpectedly long" in the legend in Supplementary Figure 1?

We re-grouped and modified Sup Fig 1 and 3 into the new **Sup. Fig 2** to clarify this point and better explain the notion of short and long 5' soft-clips.

5. There are some paragraphs and figures in the manuscript that would benefit from additional clarification:

a. Under *C. elegans* Spliced Leaders interference with direct-cDNA library preparation:

i. It is unclear which datasets were ultimately analyzed. The third sentence mentions "~1,3 million reads obtained in these initial experiments" whereas under A significant fraction of SL2 trans-spliced genes are not part of an operon, the fourth sentence mentions "~11 million reads collected in our direct cDNA sequencing". It is unclear if "initial experiments" refer to those generated using only SPP and if all three types of datasets (SPP, SL, and NP) were combined in latter analyses.

We did indeed combine all three types of experiment for the later analyses. We clarified this point in the legend of Table 1 :

"Reads from all three conditions were combined for the Spliced Leader analysis below."

and modified the mentioned text to read as follows:

"After performing three independent experiments using a Strand Switching Primer for the second strand synthesis as recommended by the Oxford Nanopore documentation. After mapping our reads onto *C. elegans* genome, we controlled their correct alignment by looking at them in Integrative Genome Viewer (IGV), and noticed an unexpected, reproducible strand bias in favor of antisense reads (Fig. 1a). Out of ~1,3 million reads obtained in these initial experiments, 95% were antisense reads."

And :

"From ~11 millions reads collected in three conditions of direct-cDNA sequencing (see Table 1),"

ii. Sentence 6 states that soft-clips are expected to contain certain elements, and so it is a little confusing to read in the next sentence that "indeed, that was the case for the minority of sense strand reads" without any further clarification for this unexpected observation. I assume only a minority of sense strand reads contained all expected elements because hairpin formation prevented complete second strand synthesis. If so, it may be helpful to the reader to allude to this.

We were trying to convey the fact that the short soft-clip of sense reads indeed contained the expected features while the more common antisense reads did not.

We modified the paragraph as follow:

The long soft-clips from antisense reads, overall, did not contain the expected features, in sharp contrast with the more conventional, but minority, sense strand reads (Sup. Fig. 2a). Additionally, we noticed that ~33% of the "long soft clips" contained a supplementary alignment matching to the sense strand of the original alignment (Sup. Fig. 2b and c).

b. Figure 5b: What percentage of full-length transcripts were identified from all reads?

The term full-length in the original manuscript was ambiguous. It is frequent to find reads that include SL sequence event if they seem to come from an internal (potentially spurious) start site. In this case the read is a full length in the sense that it captures the complete sequence of a mRNA molecule but it may not cover the entirety of a predicted isoform. Technically, all our reads containing either SL or hairpin sequences are full-length reads.

To fully take advantage of the quantitative nature of our data, we decided to focus on the most detected alignment start position for each gene to define the “main isoform” for each gene.

The new **Figure 6** does not use the term full-length to avoid any ambiguity.

Reviewer #2 (Remarks to the Author):

(...) Given that long read sequencing technology is an important tool in the characterisation of transcriptomes, the work is important to all *C. elegans* researchers using or contemplating using ONT transcriptome sequencing. However, since most, if not all, spliced leaders have this same propensity to form 5' hairpins, it is likely to be an issue for all organisms whose transcripts are substantially spliced leader *trans*-spliced. In fact, the authors could strengthen the manuscript by explicitly pointing this out.

We added the following sentence in the discussion to make this point more explicit:

“We confirmed the same strand bias in *Leishmania Tarentolae* direct-cDNA sequencing experiments²⁴, it is therefore likely that similar library behavior will extend to all species using *trans*-splicing.”

The work is clearly and beautifully presented, including a convincing mechanistic explanation for the poor sequencing quality of the sense-strand soft-clipped 5' tails. Moreover, I can independently confirm that this is a reproducible issue associated with this technology. My laboratory's own data generated using the same direct-cDNA sequencing system contains a similar degree of read bias and extended soft-clipped reads. It is gratifying to have an explanation for this issue.

One thing here, though, is that the authors are not showing any alignment statistics; it's not clear if the alignments could be improved.

We thank this reviewer for their kind words. We modified **Table 1** and added **Sup. Table 1 and 2** that indicates the alignment statistics requested.

There are two things that could help with the presentation of this issue. 1) Have the authors tried to trim and orient the reads using pyclobber?

This would have directly alerted them to the excess of VNP-VNP reads and allowed them to quantify the problem directly before alignment. 2) The reads could also have been screened for spliced leader 5' tails (and those tails then removed) before alignment.

We did attempt to use pyclobber in our 3 initial SSP runs but the software called “unusable” a majority of the reads. We elected to map all the reads to the transcriptome and realized that the large majority could indeed be mapped to a transcript (see **new Table 1**). After our *ad hoc* analysis this discrepancy is explained by the quasi systematic presence of hairpin reads that do not conform to the expected structure of FL reads (and in particular since hairpin cDNAs do not carry the SSP).

Despite these important technical issues, the depth of sequence coverage allows the authors to go on to comprehensively study *C. elegans* spliced leader usage. To some degree, this aspect of the work does not appreciably add to our current understanding of spliced leader *trans*-splicing in *C. elegans*, but as the authors state, it is the most comprehensive to date.

We thank the reviewer for his comment. To highlight this point we added the following sentence in the Discussion section:

“The data collected in this work with a few direct-cDNA sequencing run allowed us obtain a more detailed global picture of the *C. elegans* transcriptome than years of accumulated Illumina-based RNA-seq experiments. Including a quantitative characterization of spliced leader usage and the nature of non *trans*- spliced 5' extremities. This demonstrate the potential of this technology for future more targeted transcriptomics analyses.”

However, I am puzzled by the title: “A significant fraction of SL2 *trans*-spliced genes are not part of an operon”. From what is presented, the authors claim this on the basis that there are many (approximately 300) SL2 *trans*-splicing events greater 200 bp (and up to several kb) downstream of the closest upstream gene. But we know that such distances do not rule out the possibility of being in an operon, as outlined in Morton and Blumenthal, 2011 (Morton JJ, Blumenthal T. Identification of transcription start sites of *trans*-spliced genes: uncovering unusual operon arrangements. RNA. 2011;17: 327–337. See also, Blumenthal T, Davis P, Garrido-Lecca A. Operon and non-operon gene clusters in the *C. elegans* genome. WormBook. 2015; 1–20.). I would recommend that the section is re-written to consider this information and to change the sub-heading. The evidence as presented does not support the sub-title's assertion.

We replaced the section title with :

Quantitative analysis of spliced leader usage

We also replaced the categorization « Inside operon » and « outside operon » in figure 3 by : “distance from upstream gene under 200nt and over 200nt “

The most intriguing outcome of this analysis is the identification of non-*trans*-spliced transcripts that nonetheless possess inherent 5'-hairpins. This is striking and suggests the possibility that cellular adaptation to the preponderance of transcripts with 5' spliced leaders *C. elegans* has created a selective pressure that has shaped non-*trans*-spliced transcripts.

Finally, there are a couple of points raised by the Discussion that I would like to address. As noted above, the presence of hairpins in spliced leaders from other nematodes, animals and protists means that the issues documented here will apply more broadly. This should be noted and expanded upon in the discussion since it will make the impact of this work more explicit.

We added the following sentence in the discussion to make this point more explicit:

“We confirmed the same strand bias in *Leishmania Tarentolae* direct cDNA sequencing experiments²⁴, it is therefore likely that similar library behavior will extend to all species using *trans*-splicing.”

There is also a statement that requires clarification: “Additionally, while all Spliced Leaders present a modified Guanine in 5', endogenous terminal hairpins do not, which would make them uncapped mRNAs”. The authors appear to be stating that the non-spliced leader *trans*-spliced transcripts lack a cap – this is not (and cannot be) the case, they will have a standard monomethyl guanosine cap. I presume that the authors simply mean that they will not have trimethyl guanosine caps.

We have removed that sentence that was indeed poorly worded.

Minor comments and typographical errors

The prefixes in *trans/cis*-splicing should be italicised.

We modified the text accordingly

Abstract 1st line, the text states “21 nt sequence”– nematode spliced leaders are 22 nt

Indeed, SL1 adds 22nt but some variants of SL2 add only 21nt. In order to introduce the concept for nematodes and trypanosomes at large which include various different specific spliced leaders sizes we removed the inaccurate number in favor of a more general statement:

“In nematode and kinetoplastids, mRNA processing involves a trans-splicing step through which a short sequence from a snRNP replaces the original 5’ end of the primary transcript.”

“While 700 out of 1011 genes are located within 200 bp downstream...”

We modified the text accordingly

The authors use the term “messengers” at several points in the manuscript – this term is ambiguous; “mRNA” or “transcripts” would be better.

We modified the text accordingly

There are quite grammatical infelicities/errors throughout the manuscript. I detail a few below, but the authors should check through the revised manuscript carefully.

“...resulting long RNAs needs to be matured...”

“In this protists, genes do not contains introns, therefore the spliceosome’s sole function...”

We modified the text accordingly to fix the mistakes mentioned by the reviewer and performed a thorough re-read of the manuscript to catch additional typos.

Reviewer #3 (Remarks to the Author):

[Bernard et al.,] describe an interesting set of observations resulting from the application of cDNA-based long read sequencing with the oxford nanopore device. Their study suggests that RNA hairpins present in *C. elegans*’ spliced leaders (SLs) allow for self-priming during cDNA synthesis. They go on to show several examples of non-SL genes that exhibit similar hairpin structures, suggesting that this may be a widespread feature of mRNAs in the worm.

The observations are interesting and novel. I have a handful of concerns that, if remedied, would significantly strengthen the study.

Their interpretation of the species in question relies on a novel technology

1. The key step for their sense-antisense reads seems to be simple self-priming and reverse transcriptase extension. I would feel more confident in their interpretation of the reads if they could show the species existed with canonical methods. For example, RT, PCR, and sanger sequencing across a few of the most abundant hairpins is straightforward and would strengthen their conclusion as to the origin and nature of the reads. This is key as the nanopore technology is still relatively new, and thus certainly has unanticipated misinterpretations.

We have tried the strategy suggested by this reviewer of amplifying by PCR the hairpin fragment for Sanger sequencing. We designed several reverse primers that could potentially generate symmetrical amplicons centered on the hairpin loop that should provide a good matrix for sequencing. We were however not able to obtain these amplicons. In hindsight it is probably because the two strands of the hairpin, being covalently attached to each other, re-hybridize during the annealing step, thus preventing efficient polymerization from the primers.

However, our interpretation resulted from direct discussion with the technical support staff of Oxford Nanopore Technology who confirmed that the current pores behaved as we represented in Figure 1e which led the discontinuation of their "2D" reads kit that was previously relying on the reading of two strands of a hairpin (that previous version of their pores could handle).

2. The authors conclude that many SL2s are not part of operons. The key data is omitted: "We investigated...cryptic termination...but didn't find evidence (data not shown)." It is still possible the SL2s are part of operons, but the upstream RNA/polyadenylation events from whence they are derived are unstable or rare. The data presented does not speak to this possibility, and ultimately does not support their conclusion.

We replaced the section title with :

Quantitative analysis of splice leader usage

We also replaced the categorization Inside operon and outside operon in figure 3 by :
"distance from upstream gene under 200nt and over 200nt "

3. The authors suggest that many non-SL mRNAs have a terminal hairpin. The data presented is anecdotal, showing four examples (Fig 4C). It's clear they have many more non-SL mRNAs (Fig 4B). It would be helpful (and more rigorous) to have a statement about the generality of the hairpin phenomenon for the mRNAs in Fig 4B. How many of non-SL mRNAs have a recognizable hairpin? (Perhaps this is what is deemed "unidentified"/"unknown" in Fig5B, but this is not clear.)

We have clarified the meaning of « unidentified » in the modified text as follows:

In the course of this analysis we observed that for a fraction of reads that carried all the hallmarks of hairpin reads (strong antisense bias, long 5' soft-clip - see Sup. Fig. 5) the sequence of the hairpin structure was not retrieved. We labeled these "unidentified hairpins reads" as "unidentified" for short in figure 5c. We didn't find genes with a higher propensity for unidentified reads that could have indicated the absence of any 5' hairpin structure. The 3,056 genes that have only "unidentified" reads are characterized by a very low level of expression compared to the other categories (Fig 5c). It therefore seems to indicate that we are more likely confronted with a detection issue than a third mode of mRNA maturation.

4. I am apprehensive about trusting RNA folding algorithms, which will readily produce hairpins on random sequence.

The various hairpin structures presented in the figures were not derived from folding algorithms but identified through the analysis of the detected sequences in the soft-clipped region upstream of the aligned region (see figure 1c, figure 4c and 5c). We noticed the self-complementarity of the reads

on themselves in the vicinity of the end of the alignment and noticed that, as was the case for the SL reads, the last bases of the alignment were part of the stem. This internal small stem is then extended into a very long stem during the cDNA synthesis which leads our reads to have long, mostly unreadable soft-clipped sequences.

We modified the wording in the text to clarify that:

“Looking at these reads we observed that the first few nucleotides adjacent to the end of the SL1 sequence corresponded to an antisense fragment of itself followed from a portion of the sense cDNA. This reinforced the notion that the two guanine bases at the 5' extremity of the SL1 sequence had paired with two of the three cytosines in the middle of the spliced leader causing the self-priming of long hairpin cDNAs instead of the expected second strand synthesis step.”

In the Fig. 2 legend :

“Genomic location of all *C. elegans* *sls* genes and structure of their 5' hairpin strand derived from our sequence analysis”

In the Fig. 4 legend :

“For the three genes highlighted in panel b we show a partial alignment of the end of the aligned region and the beginning of the soft-clip. Arcs represent base pairing that were used to generate the hairpin models.”

The nature of the hairpin and the sense/antisense artefact is well-suited to a more rigorous test of the terminal hairpin forming behavior of mRNAs, SL or not. By examining the frequency of sense/antisense reads at a given locus, one should be able to get an accurate estimation of the terminal hairpin propensity. **For mRNAs with SLs, the sense/antisense ratio should be ~5%/95%, and for mRNAs without terminal hairpins it should be ~50%/50% (Fig 1D). This would be a rigorous way of testing the hairpin propensity of the non-SL transcripts in general.**

We have generated **supplementary Figure 8** which displays the obtained strand bias for every gene. Several outliers (n=31) could be easily identified that did not display the strong bias observed for the majority of genes in the SSP datasets but they behaved as hairpin in the other conditions.

We included the following paragraph in the Discussion:

While it remains possible that some genes produce transcripts devoid of 5' hairpins we did not find any evidence of their existence, the handful of genes that showed minimal strand bias in the SSP experiments behaved like the majority of hairpin generating genes in other experimental set up (Sup. Fig. 8). It seems therefore more likely that these genes interacted with the SSP in this set up in a similar manner as *trans*-spliced genes interacted with the SL1 primer.

5. The authors suggest that the proportion of trans-spliced mRNAs in *elegans* is higher than earlier estimates of 70% or even 85%. However, the authors provide no number. It would be informative for the authors to share their estimate of the fraction trans-spliced mRNAs.

The new **Figure 5** addresses this question directly. And we added the following paragraph in the text:

In the course of this analysis we observed that for a fraction of reads that carried all the hallmarks of hairpin reads (strong antisense bias, long 5' soft-clip - see Sup. Fig. 6) the sequence of the hairpin

structure was not retrieved. We labeled these “unidentified hairpins reads” as “unidentified” for short in Figure 5c. We didn’t find genes with a higher propensity for unidentified reads that could have indicated the absence of any 5’ hairpin structure. The 3,056 genes that have only “unidentified” reads are characterized by a very low level of expression compared to the other categories (Fig 5c). It therefore seems to indicate that is more indicative of a clack of coverage rather than an indication of a third mode of mRNA maturation.

6. The authors suggest that non-SL mRNAs are uncapped: “endogenous terminal hairpins do not, which would make them uncapped mRNAs.” My understanding is that this is inaccurate. Indeed, transcription start site mapping by others (PMID: 23260138, 23636945) has used the capped nature of non-SL mRNAs to map their 5’ends. The authors should correct this statement.

We have removed that sentence that was indeed poorly worded.

7. The axes in Fig 4A/B seem confusing and perhaps show the wrong scale. They indicate some genes have tens of thousands of nanopore reads, with some nearing 100,000. The authors libraries only have a few hundred thousand reads each, so it would seem unlikely that even a highly abundant (~1% of mRNA) would achieve such high counts. Is the axis mislabeled? Were libraries combined?

We did indeed combine all our direct-cDNA runs for the quantitative analyses. We clarified this point in the legend of Table 1 :

“Reads from all three conditions were combined for the Spliced Leader analyses below.”

8. It would be beneficial if the authors took some care to ensure typos and ambiguities in communication are removed. While I could read through most of the small typos, others left me wondering what the authors were attempting to say. For example, “...for a large number of reads the sequence of the hairpin structure was not accessible.” What does “accessible” mean in this context? Does this mean “We were unable to identify the sequence of the hairpin structure.”? Does that mean there was no hairpin structure, or that the precise position at which antisense became sense was ambiguous? I am not sure, was not able to figure it out from the rest of the section, and thus not sure what is being displayed in Figure 5B.

Indeed, we meant that the sequencing did not allow us to pinpoint the origin of the hairpin for all of the reads we obtained, even though the hallmark of hairpin cDNA reads is still present, we added Sup. Fig. 4 to address the nature of these “unidentified hairpin“ reads.

In the course of this analysis we observed that for a fraction of reads that carried all the hallmarks of hairpin reads (strong antisense bias, long 5’ soft-clip - see Sup. Fig. 6) the sequence of the hairpin structure was not retrieved. We labeled these “unidentified hairpins reads” as “unidentified” for short in figure 5c. We didn’t find genes with a higher propensity for unidentified reads that could have indicated the absence of any 5’ hairpin structure. The 3,056 genes that have only “unidentified” reads are characterized by a very low level of expression compared to the other categories (Fig 5c). It therefore seems to indicate that is more indicative of a clack of coverage rather than an indication of a third mode of mRNA maturation.

REVIEWERS' COMMENTS

Reviewer #1 (Remarks to the Author):

The authors have addressed major and minor concerns in this revised manuscript, most importantly providing information about the sequencing quality and modifying main and supplementary figures that further support the use of Nanopore direct-cDNA sequencing as a method to investigate trans-splicing. However, the authors do not convincingly address their hypothesis, stated in lines 70-72, to demonstrate that they were able to identify an even higher number of trans-spliced genes than a previous report of 85% of *C. elegans* mRNAs (line 85). Since the authors state in the abstract that mainstream transcriptome sequencing methods are unable to fully capture trans-splicing, it is crucial that the results in this manuscript demonstrate that Nanopore direct-cDNA sequencing captures more trans-splicing events than previously reported. Overall, this revised manuscript is suited for publication in Nature Communications pending additional edits.

Resolved issues:

1. It was previously unclear how the quality of the sequencing data had been assessed. The authors have addressed this issue in a new Supplementary Table 2 and Supplementary Figure 3.
2. The addition of a new Supplementary Figure 1 clarifies the distinction between short and long 5' soft-clips. Furthermore, these terms have been clearly defined in the first paragraph of the section *C. elegans* Spliced Leaders interfere with direct-cDNA library preparation under Results.
3. The new Figure 3a explores preferential usage of Spliced Leaders, however I am now puzzled by the small number of SL1 (3,780) and SL2 (1,008) genes depicted. As stated by the authors in the Introduction in lines 66-67 that early global analyses of trans-splicing estimated that 70% of *C. elegans* mRNAs are trans-spliced with newer estimates of 85%, I would have expected a much higher number of SL1 and SL2 genes than shown. Perhaps the new Figure 5 addresses this in part, but again, the number of SL genes is only 11,846.
4. All minor criticisms have been adequately addressed.

Suggested additional edits:

1. In line 56, it appears that modifying the original content has introduced additional redundancy in the sentence.
2. Sup. Figure 3b, right bar plot: For the bar on the left indicating the supplementary alignments on the same gene, should the percentage associated with the 428,034 alignments that mapped in the opposite direction be 99% ($428,034/431,749$), and not 98%?
3. I do not think Supplementary Figure 2c was discussed in the manuscript.
4. I do not think Figure 5a and Figure 5b were discussed in the manuscript.
5. There are instances where numbers mentioned in the Results are inconsistent with numbers shown in corresponding figures:
 - a. The authors state in line 136 that 40% of sense strands were recovered using the SL1 oligonucleotide instead of the supplied SSP, but the associated Figure 1d shows 27% recovery as does Supplementary Figure 1.
 - b. The authors state in line 172 that 700 out of 1,011 genes are located within 200 bp downstream of the closest upstream coding gene, but the associated Figure 3a shows 698 out of 1,008 genes.

c. The authors state in lines 212-213 that the 3,056 genes that have only unidentified reads are characterized by low expression levels, but the associated Figure 5C shows that there are 3,055 such genes.

6. For percentages noted in Figure 6, the preceding 0 values may be confusing to the reader (e.g. 03% may be mistakenly interpreted as 0.3% when it is actually 3%).

Reviewer #2 (Remarks to the Author):

I am very happy with the revised manuscript. The authors have dealt effectively with all issues from my original review. I recommend that it be accepted for publication.

Reviewer #3 (Remarks to the Author):

The inability to detect the hairpins via established methods is a finding (albeit negative) that should be included in the manuscript, if only in the text. Aside from that, the authors have largely addressed my criticisms--nice story.

Resolved issues:

1. It was previously unclear how the quality of the sequencing data had been assessed. The authors have addressed this issue in a new Supplementary Table 2 and Supplementary Figure 3.
2. The addition of a new Supplementary Figure 1 clarifies the distinction between short and long 5' soft-clips. Furthermore, these terms have been clearly defined in the first paragraph of the section C. elegans Spliced Leaders interfere with direct-cDNA library preparation under Results.
3. The new Figure 3a explores preferential usage of Spliced Leaders, however I am now puzzled by the small number of SL1 (3,780) and SL2 (1,008) genes depicted. As stated by the authors in the Introduction in lines 66-67 that early global analyses of trans-splicing estimated that 70% of C. elegans mRNAs are trans-spliced with newer estimates of 85%, I would have expected a much higher number of SL1 and SL2 genes than shown. Perhaps the new Figure 5 addresses this in part, but again, the number of SL genes is only 11,846.

We have added sentences in the corresponding section of the manuscript emphasising that the lower than expected numbers are due to a high threshold of read count necessary to perform quantitative analysis

4. All minor criticisms have been adequately addressed.

Suggested additional edits:

1. In line 56, it appears that modifying the original content has introduced additional redundancy in the sentence.
2. Sup. Figure 3b, right bar plot: For the bar on the left indicating the supplementary alignments on the same gene, should the percentage associated with the 428,034 alignments that mapped in the opposite direction be 99% (428,034/431,749), and not 98%?
3. I do not think Supplementary Figure 2c was discussed in the manuscript.
4. I do not think Figure 5a and Figure 5b were discussed in the manuscript.
5. There are instances where numbers mentioned in the Results are inconsistent with numbers shown in corresponding figures:
 - a. The authors state in line 136 that 40% of sense strands were recovered using the SL1 oligonucleotide instead of the supplied SSP, but the associated Figure 1d shows 27% recovery as does Supplementary Figure 1.
 - b. The authors state in line 172 that 700 out of 1,011 genes are located within 200 bp downstream of the closest upstream coding gene, but the associated Figure 3a shows 698 out of 1,008 genes.
 - c. The authors state in lines 212-213 that the 3,056 genes that have only unidentified reads are characterized by low expression levels, but the associated Figure 5C shows that there are 3,055 such genes.

6. For percentages noted in Figure 6, the preceding 0 values may be confusing to the reader (e.g. 03% may be mistakenly interpreted as 0.3% when it is actually 3%).

We thank the Reviewer for his keen eye. We have performed all the suggested corrections.